# A Language Anchor-Guided Method for Robust Noisy Domain Generalization

## Abstract

Real-world machine learning applications are often hindered by two critical challenges: distribution shift and label noise. Networks inherently tend to overfit to redundant, uninformative features present in the training distribution, which undermines their ability to generalize effectively to the target domain's distribution. The presence of noisy data further exacerbates this issue by inducing additional overfitting to noise, causing existing domain generalization methods to fail in effectively distinguishing invariant features from spurious ones. To address these challenges, we propose **A**nchor **A**lignment and **A**daptive **W**eighting ($A^3W$), a novel algorithm based on sample reweighting guided by natural language processing (NLP) anchors that seeks to extract representative features. In particular, $A^3W$ leverages semantic representations derived from natural language models to serve as a source of domain-invariant prior knowledge. We also introduce a weighted loss function that dynamically adjusts the contribution of each sample based on its distance to the corresponding NLP anchor, thereby improving the model's resilience to noisy labels. Extensive experiments on benchmark datasets demonstrate that $A^3W$ outperforms state-of-the-art domain generalization methods, yielding significant improvements in both accuracy and robustness across various datasets and noise levels.

## 1 Introduction

Domain Generalization (DG) has emerged as a pivotal algorithm in machine learning, aiming to develop models that can maintain high performance on previously unseen environments—or *domains*. Traditional methods often assume that training and test data share the same distribution, yet in real-world scenarios, there is frequently a substantial shift between these distributions. This phenomenon, widely referred to as *domain shift*, can cause severe performance degradation in tasks spanning computer vision, natural language processing, and medical image analysis (Wang et al., 2022). As shown in Figure 1(a)(b), even within the same class label, the distribution of feature representations can vary considerably. This variation may stem from differences in image acquisition conditions, such as lighting variations, changes in pose, or complex background environments, and even from more subtle domain-specific factors like sensor noise or camera calibration differences. Such intra-class variability poses a significant challenge for developing accurate and adaptable models, which must learn to extract invariant features that capture the true semantic essence of the class while ignoring irrelevant variations. In response, DG task has been proposed to enable robust model behavior without further fine-tuning, thereby facilitating broader applicability in diverse domains (Muandet et al., 2013; Li et al., 2018). Over time, researchers have explored various ways to align or unify source and target domains, including strategies that learn *domain-invariant* features through multi-task autoencoders, augmented architecture, or adversarial alignment (Ghifary et al., 2015; Ganin & Lempitsky, 2015; Volpi et al., 2018).

One persistent obstacle in DG is the prevalence of *spurious correlations* in deep neural networks, where models inadvertently link target labels to irrelevant, domain-specific features (Arjovsky et al., 2019; Qiao & Low, 2024). For instance, a classifier tasked with recognizing animals might rely on particular backgrounds for prediction, causing it to misclassify animals photographed in unfamiliar settings (Beery et al., 2018).

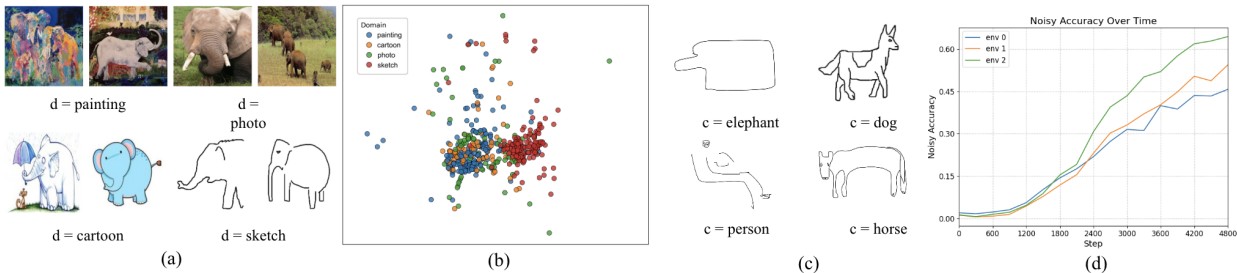

Figure 1: Illustration of domain shift using four distinct domains—painting, photo, cartoon, and sketch—for the same object class. In (a), the visual appearance of "elephant" varies substantially across domains, underscoring significant style discrepancies. In (b), the t-SNE projection shows that even for the same class, the distribution of features differs across domains, highlighting the inherent challenges of domain generalization. In (c), unclear or mislabeled samples introduces additional noise, further exacerbating the difficulty of achieving robust generalization. In (d), we show the accuracy of a network trained on noisy data: ideally, the model should resist overfitting to noise, but the graph indicates a steady increase in noise accuracy over time, suggesting progressive overfitting to noisy labels. (d) Noise accuracy, defined as the accuracy with respect to the noisy (corrupted) labels rather than the clean ground truth. Ideally, this metric should remain low if the model resists memorizing noise. The steady increase shows progressive overfitting to mislabeled data.

Such spurious correlations frequently arise because modern networks have the capacity to overfit even low-level noise, thereby anchoring their decisions on non-causal cues (Rahman et al., 2024). To address these pitfalls, researchers have proposed a variety of techniques, from causal inference algorithms like Invariant Risk Minimization (IRM) (Arjovsky et al., 2019) to methods specifically designed to disentangle spurious and invariant features (Wang et al., 2024). For instance, Discover and Cure (DISC) Wu et al. (2023) strategy was introduced to detect suspicious features and systematically mitigate their impact by leveraging interpretable domain knowledge. The development of methods rooted in causal representation learning also arises to mitigate the reliance on spurious correlations (Lv et al., 2022). Despite these advances, achieving robust generalization under significant domain shift and considerable label noise persists as an open research challenge that necessitates more comprehensive methods and better theoretical insights.

Existing domain generalization (DG) approaches have predominantly focused on aligning feature distributions or disentangling spurious factors using data-driven methods (Wang et al., 2022). While these techniques have paved the way for significant advances, they often encounter considerable difficulties when label noise is prevalent or when data lack distinct domain-invariant cues (Nigam et al., 2020), leading to rapid degradation in network performance as noise intensity increases. This challenge is further exacerbated in large, over-parameterized networks that are prone to memorizing spurious correlations, thereby amplifying errors. Indeed, estimates suggest that 8% to 38.5% of real-world data may suffer from label corruption, undermining the generalizability of models that fail to distinguish genuine signals from incidental patterns (Song et al., 2022a). Although some DG algorithms inherently exhibit robustness to label noise, in general cases they lack performance superiority compared to empirical risk minimization (ERM) baseline (Qiao & Low, 2024). Consequently, understanding how to boost robustness against both domain shifts and noisy labels remains a pressing goal in building truly reliable artificial intelligence systems.

In light of these challenges, we consider an alternative approach that focuses on introducing novel sources of features from diverse perspectives to guide the learning pipeline. An emerging line of research has investigated the role of *external knowledge*, which can guide models toward more semantically grounded representations (Dash et al., 2022). Some studies incorporate domain knowledge directly into the learning pipeline, for instance by embedding specialized scientific insights into feature extraction or loss design (Von Rueden et al., 2021). In other cases, researchers have shown that domain knowledge can significantly enhance model interpretability and robustness. For example, in laser-induced breakdown spectroscopy, integrated expert knowledge improved quantification performance (Song et al., 2022b). Another promising direction involves multi-modal alignment, in which semantic text descriptions serve as stable anchors that help networks transcend purely visual or sensor-driven biases (Liu & Wang, 2023). Models like CLIP (Radford et al., 2021)

exemplify this principle by projecting both text and images into a shared embedding space, improving generalization across dissimilar environments. Yet critical barriers remain regarding how to ensure that the guidance remains valid across diverse domains and how to structure knowledge encoding effectively to align with modern deep learning architectures (Dash et al., 2022).

Inspired by this recent trend of knowledge integration, we propose a new algorithm called **A**nchor **A**lignment and **A**daptive **W**eighting ($A^3W$), which integrates external knowledge—specifically, linguistic cues derived from large-scale language models—into the DG pipeline. Our central insight is that *NLP anchors*, such as text embeddings from CLIP or analogous language models, provide *domain-invariant* references capable of steering learned representations away from spurious and noisy cues. By aligning intermediate feature spaces with these carefully chosen textual anchors, $A^3W$ encourages semantic consistency while limiting the influence of mislabeled or outlier samples. Moreover, we introduce a *weighted loss function* that dynamically modulates each sample's impact based on its proximity to an anchor, thereby enhancing robustness to label noise. We validate our method through extensive experiments on multiple DG benchmarks, confirming that $A^3W$ surpasses existing approaches in terms of both accuracy and resilience to shifting and noisy data. Our contributions can be summarized as follows:

- We propose an iterative update algorithm that unifies external semantic knowledge and image-based features, enabling more robust and interpretable generalization across unseen domains.

- We address the emerging and challenging problem of domain generalization under noisy data by introducing NLP anchors derived from large-scale language models (e.g., CLIP), which provide domain-invariant and semantically rich constraints. Furthermore, we design a novel *weighted loss function* that dynamically adjusts each sample's contribution based on its distance to the corresponding anchor, emphasizing reliable samples and down-weighting noisy outliers.

- We conduct extensive experiments on multiple domain generalization benchmarks to demonstrate that $A^3W$ outperforms state-of-the-art methods in terms of accuracy, robustness, and adaptability.

## 2 Related Work

### 2.1 Domain Generalization

Domain generalization (DG) addresses the challenge of training models that can perform well on previously unseen domains, without access to labeled examples from those domains at training time. A significant challenge in DG is the presence of *spurious correlations* in deep neural networks, which hinders generalization across diverse settings (Arjovsky et al., 2019). These correlations arise when models inadvertently rely on domain-specific artifacts (i.e. features that are incidentally correlated with the target labels in the training data) rather than on the truly invariant properties that are essential for robust performance. Consequently, when models encounter data from an unseen domain, their over-reliance on these spurious cues leads to performance degradation when deployed in unseen domains. Early methods focused on learning domain-invariant representations through shallow or deep feature alignments, aiming to eliminate domain-specific information while preserving task-relevant features. For instance, Muandet et al. (2013) propose Domain-Invariant Component Analysis (DICA), a kernel-based method to learn invariant features by minimizing the discrepancy across source domains. Ghifary et al. (2015) use multi-task autoencoders to learn generic representations.

More recent works resort to data augmentation or adversarial strategies to synthesize novel training distributions, thereby exposing models to a richer variety of samples. Volpi et al. (2018) introduced adversarial data augmentation by generating worst-case perturbations to expose the model to a broader set of variations during training to improve its ability to handle unseen shifts. Ganin & Lempitsky (2015) introduced domain-adversarial training using gradient reversal layers to align feature distributions. Li et al. (2018) explored meta-learning strategies to enhance DG by simulating domain shifts during training. Shankar et al. (2018) proposed a generalization method using domain-specific perturbations to improve robustness. Another class of augmentation-based methods applies Mixup strategies, interpolating data points across domains to improve robustness (Yan et al., 2020). Style transfer methods have also been introduced to augment training

data with diverse visual styles, improving out-of-distribution performance (Nam et al., 2021). Carlucci et al. (2019) proposed a self-supervised approach that solves jigsaw puzzles to learn more generalized features.

Beyond augmentation, several methods have turned their focus to regularization to improve generalization. Balaji et al. (2018) introduced a regularization algorithm by learning a regularizer modeling the objective of DG that a model trained on one domain to generalize effectively. Dou et al. (2019) utilized episodic training to simulate domain shifts and improve generalization. Zhou et al. (2021) introduced a regularization approach based on domain-adaptive ensemble learning to enhance a model's ability to generalize across different domains. Proxy-based Contrastive Learning (PCL) (Du et al., 2021) leverages class-specific proxies to improve domain generalization through contrastive alignment. Our approach is conceptually related in that it also employs class-level references ("anchors"), but differs by grounding them in external language models and combining them with adaptive weighting for noise robustness. Furthermore, recent research has revisited the issue of spurious correlations in DG, proposing methods to mitigate their impact. Qin et al. (2024) explored building a structural causal model for representation learning to address spurious correlations. Ma et al. (2024) introduced FedCD, a federated domain generalization algorithm that employs a spurious correlation intervener for self-supervised feature intervention and a risk extrapolation aggregation strategy to reduce reliance on misleading shortcuts and boost performance on unseen domains. More recently, multimodal approaches have also emerged; for instance, Liu & Wang (2023) proposed Text-guided Domain Generalization (TDG), which enriches domain information through generated domain-relevant text features combined with CLIP. Similarly, Zhang et al. (2023) introduced method that distills CLIP with language guidance for DG, showing that language-derived embeddings can substantially improve robustness. While related in its use of textual guidance, TDG and language-guided distillation primarily focus on domain diversity, whereas our method additionally incorporates class-specific anchors and adaptive weighting to explicitly handle noisy labels alongside domain shift.

## 2.2 Learning under noisy labels

Learning under noisy labels addresses the challenge of training models when annotated data contain errors or inconsistencies. Such noise often arises in real-world scenarios due to human annotation mistakes, weak labeling processes, or automated data collection pipelines. When training with noisy labels, deep neural networks can overfit to incorrect annotations, resulting in degraded performance and reduced robustness (Song et al., 2022a). Initial methods focused on designing robust loss functions and label-correction strategies to mitigate the impact of label noise. For example, Reed et al. (2014) introduced a "bootstrapping" approach that combines model predictions with the (potentially noisy) labels to guide learning. Goldberger & Ben-Reuven (2017) proposed adding a noise adaptation layer to model the corruption process directly. Veit et al. (2017) leveraged a small clean dataset to estimate noise statistics and correct the loss for mislabeled examples. These methods laid the groundwork for more advanced noisy-label handling techniques.

Subsequent research explored "co-teaching" strategies to further reduce the impact of noisy annotations. Han et al. (2018) proposed a co-teaching algorithm wherein two networks train simultaneously, exchanging likely clean samples to avoid memorizing label noise. MentorNet (Jiang et al., 2018) introduced a curriculum-based approach, using a "mentor" network to select reliable samples for the "student" network, progressively ignoring data points suspected to be noisy. A complementary direction has focused on iterative label refinement. Tanaka et al. (2018) presented a joint optimization algorithm that updates labels alongside network parameters, gradually reducing noise in the dataset. To improve robustness further, recent efforts incorporate data augmentation strategies specifically tailored for noisy labels. For instance, Nishi et al. (2021) systematically studied augmentation techniques to enhance the network's tolerance to corrupted annotations.

Recent studies delve deeper into noisy-label learning in conjunction with large-scale or real-world datasets. DivideMix (Li et al., 2020) treated noisy-label learning as a semi-supervised problem, splitting data dynamically into clean and noisy sets for more targeted training. Song et al. (2022a) provided a comprehensive survey on deep learning with noisy labels, highlighting emerging trends such as instance-dependent noise modeling and robust early-learning regularization. These approaches emphasize balancing label correction, sample selection, and model regularization. Robust learning under noisy labels has become increasingly important for domain generalization tasks, as noisy annotations can exacerbate spurious correlations and hinder out-of-domain performance. Consequently, integrating advanced noisy-label handling into DG pipelines re-

mains an active area of research. Some recent research has begun to address the intersection of label noise and domain shifts. Qiao & Low (2024) indicated that while label noise can sometimes serve as a form of regularization, in scenarios with significant domain shifts it may reinforce spurious correlations, ultimately degrading performance on unseen domains. These findings underscore the need for integrated approaches that *jointly mitigate label noise and domain shifts* to achieve robust out-of-distribution generalization.

## 3 Method

### 3.1 Problem Formulation

Let $\mathcal{D}_S$ denote a set of source domains, and $\mathcal{D}_T$ denote the target domain. We assume that $\mathcal{D}_S$ and $\mathcal{D}_T$ share the input space $\mathcal{X}$ and label space $\mathcal{Y}$, in which each domain $d \in \{1, \ldots, \mathcal{D}\}$ has data drawn from a joint distribution $P_d(X, Y)$ where $X \in \mathcal{X}$ and $Y \in \mathcal{Y}$. In practice, the observed labels are corrupted by noise, i.e. for each sample with true label $y$, we observe a noisy label $\tilde{y}$ generated by:

$$\tilde{y} = \begin{cases} y, & \text{with probability } 1 - p, \\ \tilde{y}' \sim Q(\cdot \mid y), & \text{with probability } p, \end{cases}$$

where $p$ is the probability that the label is corrupted, and $Q(\tilde{y} \mid y)$ is a conditional distribution over possible noisy labels given $y$. For a classification problem with $C$ classes with $c \in \{1, \ldots, C\}$, this means that the true label is flipped to a different class with probability $p$.

*Noisy Domain Generalization:* The proposed method aims to generalize to unseen domains while preventing overfitting to noisy data. In this setting, the training dataset $\mathcal{S}_{tr}$ is contaminated with label noise. We assume that the testing dataset $\mathcal{S}_{te}$ is drawn from a clean distribution with labels $\mathcal{Y}_{te}$. Our objective is to train a model on the noisy training dataset $\mathcal{S}_{tr}$ that learns robust and domain-invariant representations, thereby minimizing the classification error on the clean target domain $\mathcal{D}_{te}$.

### 3.2 Motivation

**Domain generalization is a challenging problem primarily due to the tendency of networks to learn spurious correlations between features and labels.** Such correlations can cause overfitting, even in tasks where the network achieves high performance on the training data. Observations from several works in DG highlighted the challenge for standard networks to learn representations that are robust or invariant enough to generalize well to unseen domains. The DomainNet (Peng et al., 2019) benchmark revealed that many current models struggle to transfer knowledge effectively across heterogeneous data sources. Gulrajani & Lopez-Paz (2020) showed that numerous DG algorithms fail to significantly outperform a simple empirical risk minimization (ERM) baseline. Salaudeen & Koyejo (2022) demonstrated that incomplete constraints can lead to suboptimal generalization performance because the network may not fully disentangle invariant from spurious features. On the other hand, Ben-David et al. (2010) in their work proved that while many methods can enforce domain invariance on the training domains, this invariance sometimes comes at the expense of losing the discriminative power of the features in unseen domains.

**Natural occurrences of noise in datasets further exacerbated network overfitting.** Noise can arise from a variety of sources, such as sensor inaccuracies, annotation errors, and environmental variations. Figure 1(c) presents examples of real-world noise. The figure displays sample images from the PACS (Li et al., 2017) dataset, where some images are either incorrectly annotated or are ambiguous and unclear even to the human eye. As of current DG algorithms, Qiao & Low (2024) has proved that DG algorithms inherently provide label-noise robustness; however, our observations indicate that performance still drops significantly as noise levels increase. Figure 1(d) illustrates the impact of noise on network fitting. Here, we define noise accuracy as the classification accuracy of the model against the corrupted labels, which quantifies the extent to which the network memorizes noise instead of learning robust features. Ideally, a network trained on noisy data would exhibit relatively stable accuracy throughout training iterations, indicating resistance to overfitting. However, the sharp increase in accuracy across all domains as training progresses suggests that the network is overfitting to the noise. This observation led us to rethink an alternative way

to provide more guidance to the feature extraction process for learning better representation. Inspired by works on leveraging external knowledge for classification to bridge representation from different modalities, we propose to harness the power of semantic embeddings to provide systematic guidance during training. In our approach, we incorporate semantic guidance to weight samples based on their consistency with predefined class prototypes or anchors, allowing us to mitigate the impact of data that exhibit significant noise or deviate from the primary training distribution. Formally, a semantic anchor $\mathbf{a_c} \in \mathbb{R}^m$ is computed for each class $c$, and the goal is to encourage the feature extractor $f : \mathcal{X} \to \mathbb{R}^m$ to map an input $\mathbf{x}$ to a representation $f(\mathbf{x})$ that aligns with the corresponding anchor $\mathbf{a_{\tilde{y}}}$ even in the presence of noise.

### 3.3   Algorithm Overview for $A^3W$

To address the challenges above, the algorithm of $A^3W$ is designed to learn a robust feature representation and classifier that generalizes well to the unseen target domain $\mathcal{D}_T$, despite the presence of noisy labels and distribution shifts across domains. At its core, $A^3W$ aims to align the projected features extracted from input images with fixed semantic anchors that act as dependable sources of knowledge. Furthermore, rather than employing a binary selection mechanism which can discard valuable information, our approach uses a continuous weighting scheme that assigns each sample an importance value, effectively preserving the gradient contributions from all samples and enabling smoother, more adaptive optimization. The overall architecture is shown in Figure 2, where we first obtain the natural language processing (NLP) anchors, and then use them to iteratively guide the updates for the featurizer, classifier, and projection layers. These stages ensure that the model learns domain-invariant representations while maintaining stable optimization. The process mainly consists of three steps, with steps 2-3 being iterative:

*1) NLP anchor setting phase*: we start by computing semantic anchors using a pretrained language-vision model (i.e., CLIP) as text encoder. For each class $c$, a text prompt is generated using a template, which is tokenized and encoded by the CLIP model. The anchors $\mathbf{\tilde{a}_c}$ are then stacked into an anchor matrix and are used to initialize a set of linear projectors $\{Proj_c\}_{c=1}^C$ that map image features into the same semantic space.

*2) Main Model Update*: we subsequently update the featurizer and classifier to improve classification performance using the combined loss of the weighted alignment and cross-entropy. During **inference**, the exponential moving average (EMA) network, initialized as a deep copy of the primary network, is employed to produce predictions, as it generally yields more stable and robust outputs compared to the primary network.

*3) Mapping Layer Optimization*: Each class is assigned a trainable mapping layer $Proj_c$, which projects feature embeddings into the NLP anchor space. The mapping layers are updated iteratively using our weighted loss function.

### 3.4   Obtaining NLP Anchors via CLIP Convention

In our method, the text encoder $T(\cdot)$ is based on a transformer architecture, as used in the CLIP model. For a given class $c$, we first construct a text prompt by appending a fixed template to the class name, and then encode this prompt using a pretrained text encoder $T(\cdot)$ to obtain an unnormalized anchor $\mathbf{a_c}$. To ensure that the anchor lies on the unit hypersphere, we normalize it, yielding $\mathbf{\tilde{a}_c}$. The formulation is given by:

$$Prompt(c) = \texttt{"a photo of a } c\texttt{"}$$
$$a_c = T\big(Prompt(c)\big) \tag{1}$$

The process begins by converting a given text prompt $Prompt(c)$ into a sequence of discrete tokens using a tokenizer. Let $t = [t_1, t_2, \ldots, t_L]$ denote the resulting token sequence, where $L$ is the length of the sequence. Each token $t_i$ is then mapped to a continuous vector via an embedding matrix $\mathbf{E} \in \mathbb{R}^{V \times m}$, where $V$ is the vocabulary size and $d$ is the embedding dimension:

$$\mathbf{e_i} = \mathbf{E}(t_i), \quad \text{for } i = 1, \ldots, L. \tag{2}$$

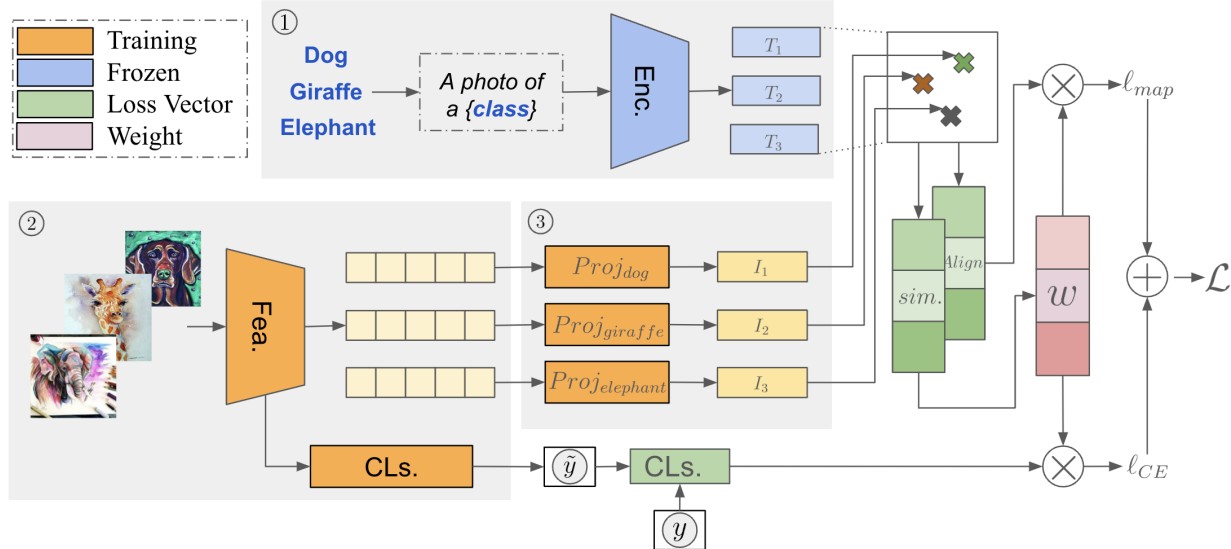

Figure 2: Architecture of $A^3W$. This diagram illustrates the end-to-end workflow and key components of $A^3W$. The encoder (Enc.) converts input text into embeddings, which are then refined by the featurizer (Fea.) into a more informative representation. The classifier (CLs.) leverages these refined features for prediction. Additionally, the similarity module (sim.) computes the cosine similarity between the embedding anchor and the projected features, while the alignment module (align.) creates deep copies of this similarity for weight ($w$) computation.

These token embeddings are augmented with positional encodings ($\mathbf{p}$) to incorporate the order of the tokens, resulting in a sequence of enriched embeddings:

$$\tilde{\mathbf{e}}_{\mathbf{i}} = \mathbf{e}_{\mathbf{i}} + \mathbf{p}_{\mathbf{i}}, \quad \text{for } i = 1, \ldots, L, \tag{3}$$

where $p_i$ is the positional encoding for the $i$-th token. The sequence $\{\tilde{e}_1, \tilde{e}_2, \ldots, \tilde{e}_L\}$ is then fed into a transformer encoder, which consists of multiple layers of self-attention and feedforward networks. This produces a set of contextualized token representations:

$$\mathbf{H} = \text{Transformer}\big([\tilde{\mathbf{e}}_{\mathbf{1}}, \tilde{\mathbf{e}}_{\mathbf{2}}, \ldots, \tilde{\mathbf{e}}_{\mathbf{L}}]\big) = [\mathbf{h}_{\mathbf{1}}, \mathbf{h}_{\mathbf{2}}, \ldots, \mathbf{h}_{\mathbf{L}}]. \tag{4}$$

Typically, a special token (such as $[EOS]$ or $[CLS]$) is appended to the input sequence, and its corresponding output $h_c$ is used as the aggregate representation of the entire text prompt. Finally, a learned linear projection ($W_{proj}$) is applied to obtain the final text embedding:

$$\mathbf{a}_{\mathbf{c}} = \mathbf{W}_{\text{proj}} \, \mathbf{h}_{\mathbf{c}}, \tag{5}$$

which is then normalized to ensure that it lies on the unit hypersphere:

$$\tilde{\mathbf{a}}_{\mathbf{c}} = \frac{\mathbf{a}_{\mathbf{c}}}{\|\mathbf{a}_{\mathbf{c}}\|}. \tag{6}$$

which serves as the NLP anchor for class $c$, providing a robust semantic reference that guides the image feature extraction process.

### 3.5 Warm-Up Training

The warm-up training phase occupies the first 10% of the total training steps and is designed to stabilize the network before introducing more complex loss terms, such as the alignment loss. During this phase, the featurizer $f$ and classifier $g$ is updated with cross-entropy loss $\ell_{CE}$. At each training step, a minibatch of samples is obtained from the dataset. The inputs $\mathbf{x_i}$ are first concatenated and passed through the feature

extractor $f(\cdot)$, implemented using a ResNet-based architecture (He et al., 2016), to generate latent features. These features are then fed into the classifier $g(\cdot)$ to produce predictions. The classifier is optimized using stochastic gradient descent by minimizing the cross-entropy loss defined as

$$\mathcal{L}_{\text{warm-up}} = \sum_{i=1}^{N} \ell_{CE}\Big(g\big(f(\mathbf{x_i})\big), y_i\Big), \tag{7}$$

where $y_i$ are the true labels and $N$ is the number of samples in the minibatch. In parallel, the moving average network is initialized as a deep copy of the primary network and is updated iteratively to stabilize learning. Specifically, after a predefined number of iterations, the EMA network parameters $\boldsymbol{\theta}_{\text{ema}}$ are updated as follows:

$$\boldsymbol{\theta}_{\text{ema}} \leftarrow \frac{\boldsymbol{\theta}_{\text{ema}} \cdot \text{ema\_count} + \boldsymbol{\theta}}{\text{ema\_count} + 1}, \tag{8}$$

where $\boldsymbol{\theta}$ are the parameters of the primary network and ema_count tracks the number of updates. After each update, the EMA network is refreshed according to the update rule described above.

### 3.6 Weighted Loss

Let $\{(\mathbf{x}_i, y_i)\}_{i=1}^{N}$ denote a batch of $N$ training samples, where $\mathbf{x}_i$ is the $i$th input feature vector and $y_i \in \{1, 2, \ldots, C\}$ is its corresponding class label for a total of $C$ classes. We denote by $Proj_{y_i}(\cdot)$ the mapping layer corresponding to label $y_i$ that projects feature embeddings into a semantic space aligned with NLP-derived anchors, and by $\lambda$ a hyperparameter balancing the alignment loss and the classification loss.

*Alignment Loss:* One way to enforce that the features align with the corresponding NLP anchor is to minimize the negative cosine similarity loss between the mapped feature representation $Proj_{y_i}(f(\mathbf{x}_i))$ and the fixed NLP anchor $\mathbf{a}_{y_i}$. This loss is defined as

$$\mathcal{L}_{\text{anchor}}(\mathbf{x}_i) = -\frac{Proj_{y_i}(f(\mathbf{x}_i)) \cdot \mathbf{a}_{y_i}}{\|Proj_{y_i}(f(\mathbf{x}_i))\| \, \|\mathbf{a}_{y_i}\|}. \tag{9}$$

By minimizing $\mathcal{L}_{\text{anchor}}$, we effectively maximizes the cosine similarity between the projected features and the NLP anchor. This process ensures that the learned features are directionally aligned with the semantic anchors, ultimately promoting representations that are both semantically meaningful and domain-invariant.

*Weight Computation:* To prioritize high-confidence training instances, we assign importance weights to samples based on their similarity to the NLP anchors. We introduce a temperature parameter $\tau$ to adjust the sharpness of the softmax weighting distribution—higher values of $\tau$ yield a more peaked distribution that emphasizes strongly aligned samples, whereas lower values produce a softer weighting. Specifically, we first compute the softmax weights over the alignment costs:

$$w_i = \frac{\exp\Big(-\tau L_i\Big)}{\sum_{j=1}^{N} \exp\Big(-\tau L_j\Big)}, \tag{10}$$

where $L_i = -\cos\Big(Proj_{y_i}(f(\mathbf{x}_i)), \mathbf{a}_{y_i}\Big)$. Rewriting this, we obtain:

$$w_i = \frac{\exp\Big(\tau \cos\big(Proj_{y_i}(f(\mathbf{x}_i)), \mathbf{a}_{y_i}\big)\Big)}{\sum_{j=1}^{N} \exp\Big(\tau \cos\big(Proj_{y_j}(f(\mathbf{x}_j)), \mathbf{a}_{y_j}\big)\Big)}. \tag{11}$$

*Overall Loss:* To maintain classification accuracy while leveraging the selective sampling, we incorporate a weighted cross-entropy loss, where samples with higher $w_i$ values contribute more significantly to the overall loss. The final loss function combines the alignment loss and the weighted cross-entropy loss:

$$\mathcal{L} = \lambda \sum_{i=1}^{N} w_i \Big[-\cos\big(Proj_{y_i}(f(\mathbf{x}_i)), \mathbf{a}_{y_i}\big)\Big] + \sum_{i=1}^{N} w_i \, \ell_{CE}\Big(g\big(f(\mathbf{x}_i)\big), y_i\Big) \tag{12}$$

The overall loss function serves a dual purpose. The alignment loss minimizes the negative cosine similarity between the projected features and their corresponding NLP anchors, compelling the model to learn representations that are semantically consistent with robust class prototypes. This semantic alignment acts as a natural filter for noise, since noisy or out-of-distribution samples tend to have representations that poorly align with their class anchors. Consequently, when the cross-entropy loss minimizes classification error, it focuses primarily on high-confidence samples that are well-aligned. Thus, by jointly minimizing both alignment and classification errors, the loss function not only ensures accurate predictions but also inherently suppresses the adverse influence of noisy data.

Reformulating the equation above, we obtain

$$\mathcal{L} \;=\; \sum_{i=1}^{N} w_i \left[ \lambda \left( -\cos\big(Proj_{y_i}(f(\mathbf{x}_i)), \mathbf{a}_{y_i}\big) \right) \;+\; \ell_{CE}\big(g(f(\mathbf{x}_i)), y_i\big) \right] \tag{13}$$

From this formulation, the weight term $w_i$ acts as an importance factor that regulates the influence of each sample on the overall loss. This regularization mechanism ensures that both the alignment loss and the classification loss emphasize reliable, high-confidence data. By down-weighting the contribution of samples that are likely noisy or out-of-distribution, the weight term effectively minimizes the adverse impact of noise during training. Consequently, this leads to more robust feature learning and improved classification accuracy.

### 3.7 Iterative Update

After the warm-up, the full $A^3W$ optimization procedure begins. We alternate between updating 1) the featurizer and classifier to improve classification performance; 2) the mapping layers are only updated periodically to prevent overfitting and to maintain stability. Specifically, the mapping layers are updated once every 10% of the total training steps to avoid overfitting and maintain stability. Let step denote the current training iteration, steps_per_epoch the number of iterations in one epoch, and $n_{\text{steps}}$ the total number of iterations. The condition for updating the mapping layers is:

$$\left( \frac{\text{step}}{\text{steps\_per\_epoch}} \right) \quad \text{mod} \quad \left( \frac{\text{n\_steps}}{\text{steps\_per\_epoch} \times 10} \right) = 0. \tag{14}$$

which ensures that the mapping layers are updated once every 10% of the total training steps. The update loss uses:

$$\mathcal{L} = \lambda \sum_{i=1}^{N} \mathbf{w_i} L_i + \sum_{i=1}^{N} \mathbf{w_i} \ell_{CE}(g(f(\mathbf{x_i})), y_i), \tag{15}$$

where $w_i$ are softmax-scaled importance weights presented above. The training procedure is summarized in Algorithm 1. Each training step consists of sampling a mini-batch from the dataset, computing both the alignment loss and weighted cross-entropy loss, and updating either the mapping layers or the featurizer and classifier based on the step schedule. The algorithm continues until convergence or until the predefined number of steps is reached.

### 3.8 Theoretical Insights

Our algorithm is designed to mitigate the adverse effects of label noise and domain shifts by integrating several key components, as formalized by our previously defined equations. Our goal is to learn a predictor $h\colon \mathcal{X} \to \mathcal{Y}$ that minimizes the error on $\mathcal{D}_T$. Define the *expected error* of $h$ on domain $\mathcal{D}_i$ as

$$\epsilon_{P_d}(h) \;=\; \mathbb{E}_{(x,y)\sim P_d}\big[\mathbf{1}\{h(x) \neq y\}\big], \tag{16}$$

where $P_d$ denotes the joint probability distribution over the input space $\mathcal{X}$ and label space $\mathcal{Y}$ for domain $\mathcal{D}_i$, capturing both the inherent variability of the data and the process by which labels may be corrupted (i.e., $y$ may be flipped to $\tilde{y}$ with probability $p$). When $p$ is high, standard empirical risk minimization can easily

---

**Algorithm 1** Training Outline for $A^3W$

---

**Require:** Dataset $\mathcal{D}$ with classes $\{1, \ldots, C\}$, hyperparameters $\lambda, \tau, \ldots$
 1: **Initialize:** featurizer $f(\cdot)$, classifier $g(\cdot)$, empty mapping layers $\{Proj_1, \ldots, Proj_C\}$
 2: **Set NLP anchors:** $\{\mathbf{a_1}, \ldots, \mathbf{a_C}\}$ via CLIP (Algorithm invokes set_nlp_anchor)
 3: **Warm-up Training:**
 4: **for** 10% of steps **do**
 5:     Sample mini-batch $\{(\mathbf{x_i}, y_i)\}$ from $\mathcal{D}$
 6:     Update parameters with $\mathcal{L}_{\text{warm-up}}$
 7: **end for**
 8: **Main Training:**
 9: **for** step $= 1$ to n_steps **do**
10:     Sample mini-batch $\{(\mathbf{x_i}, y_i)\}$ from $\mathcal{D}$
11:     **if** *condition for maplayer update is met* **then**
12:         Update mapping layers with $\mathcal{L}$
13:     **else**
14:         Update featurizer and classifier and layers with $\mathcal{L}$
15:     **end if**
16: **end for**

---

overfit to spurious correlations in the noisy labels. To counteract this, the alignment loss (see Eq. 9) serves as a semantic prior that constrains the feature extractor $f$ to produce representations that lie close to the fixed semantic anchors. This regularizing effect of the alignment loss can be formalized as follows:

**Lemma 3.1** (Semantic Prior Restricts Hypothesis Space)**.** *Suppose $\mathcal{H}$ is the space of predictors induced by $(f, \{Proj_c\})$. If $\max_{x,c} \|\nabla \mathcal{L}_{anchor}(x, c)\| \leq \gamma$, then $\mathcal{H}$ excludes functions whose representations deviate from the anchors by more than a constant factor related to $\gamma$. In particular, spurious correlations that push the embeddings away from these anchors become suboptimal.*

*Proof.* By definition,

$$\nabla \mathcal{L}_{\text{anchor}}(x, c) = -\nabla \cos\left(Proj_c(f(x)), \mathbf{a_c}\right). \tag{17}$$

A uniformly bounded gradient implies that local changes in $f(x)$ away from $\mathbf{a}_c$ incur non-negligible costs. Hence, any hypothesis that aligns poorly with anchors sees a high penalty, effectively restricting the feasible region of $\mathcal{H}$. □

Lemma 3.1 shows that anchor alignment behaves like a regularizer, steering the network away from memorizing noise-laden features. To further enhance robustness, we employ a continuous weighting scheme (Eq. 11) that assigns higher importance to samples with strong semantic alignment. This reweighting mechanism effectively adjusts the empirical risk, as captured in our overall loss function (Eq. 12), so that samples likely to be correctly labeled have a greater influence during training.

**Theorem 3.1** (Robustness under Weighted ERM)**.** *Let the learned hypothesis $h\colon \mathcal{X} \to \mathcal{Y}$ be defined as*

$$h(x) = g(f(x)), \tag{18}$$

*where $f$ is the feature extractor and $g$ is the classifier. Suppose that a fraction $\alpha$ of the training samples in domain $\mathcal{D}_i$ are corrupted. Then, if the temperature parameter $\tau$ is sufficiently large, the softmax weights $w_i$ computed via Eq. 11 will concentrate on the uncorrupted samples. This concentration effectively reweights the empirical distribution to approximate a clean distribution, such that the overall risk via Eq. 12 approximates the risk on noise-free data. Consequently, the learned hypothesis $h$ is less prone to overfitting to label noise and spurious correlations.*

*Proof.* When $\tau$ is large, the exponential function in Eq. 11 amplifies differences in the alignment cost. If a corrupted sample $(x_i, \tilde{y}_i)$ has an incorrect label, the alignment cost $\mathcal{L}_{\text{anchor}}(x_i, \tilde{y}_i)$ tends to be higher (poorer

alignment). Hence, $w_i$ becomes small. This effectively filters out mislabeled samples from dominating the training objective, approximating the scenario of training on mostly correct labels. □

Theorem 3.1 indicates that our weighting strategy can mitigate the detrimental effects of noise, enabling the model to approximate the true (clean) distribution more closely. These results not only demonstrate the efficacy of our reweighting strategy in handling noisy labels but also motivate a further examination of how our approach reduces the discrepancy between noisy and clean distributions. Let $P$ and $Q$ be the distributions of the source and target domains, respectively, with label noise in $P$. Many domain generalization results rely on bounding a distributional divergence $\text{div}(P, Q)$. Noise can inflate this divergence by altering label proportions or feature-label mappings. However, semantic alignment and selective reweighting encourage the model to focus on consistent semantic cues, thereby lowering the *effective* divergence to the clean distribution.

Formally, define the weighted empirical distribution

$$\widehat{P} = \sum_{i=1}^{N} w_i \delta_{(x_i, y_i)}, \tag{19}$$

where $\delta$ denotes the Dirac measure and each sample $(x_i, y_i)$ is reweighted by its importance $w_i$. In other words, instead of treating each sample equally as in standard empirical risk minimization, we solve:

$$\min_{h \in \mathcal{H}} \mathbb{E}_{(x,y) \sim \widehat{P}}\big[\ell\big(h(x), y\big)\big] = \min_{h \in \mathcal{H}} \sum_{i=1}^{N} w_i \, \ell\big(h(x_i), y_i\big). \tag{20}$$

This reweighting mechanism effectively suppresses the influence of corrupted samples in the empirical risk minimization process. As a result, the reweighted distribution $\widehat{P}$ becomes a closer approximation of the noise-free (clean) distribution. Formally, by minimizing the risk under $\widehat{P}$, the learned hypothesis $h$ is more likely to reflect the patterns present in the clean data rather than the spurious correlations induced by noise. In other words, by down-weighting noisy samples, our approach reduces the divergence between the empirical distribution $\widehat{P}$ and the true clean distribution $Q$, i.e., $\text{div}(\widehat{P}, Q)$ is reduced. This reduction in discrepancy implies that the model is trained on a distribution that better represents the underlying data, ultimately leading to improved generalization performance on the unseen target domain.

**Theorem 3.2** (Generalization bound under discrepancy specialized to $A^3W$). *Let $\ell_{CE}$ be cross-entropy loss with softmax logits. Assume features and anchors are $\ell_2$-normalized, and classifier weights are bounded so that logits satisfy $\|z\| \leq B$. Then $\ell_{CE}$ is 1-Lipschitz in $z$. For any $h = g \circ f$ with featurizer $f$ and classifier $g$, we have*

$$R_Q(h) \leq R_{\widehat{P}_w}(h) + \text{disc}_{\ell_{CE}}(f\#\widehat{P}_w, f\#Q) + \lambda^\star + \mathcal{E}_n, \tag{21}$$

*where $\widehat{P}_w$ is the $A^3W$ reweighted distribution, $\lambda^\star = \inf_{h \in \mathcal{H}}(R_{P^\star}(h) + R_Q(h))$ for the clean source $P^\star$, and $\mathcal{E}_n$ is a standard sample-complexity term scaling with the effective sample size $n_{\text{eff}} = 1/\sum_i w_i^2$.*

**Corollary 3.1** (Wasserstein instantiation). *If $f$ is $L_f$-Lipschitz and $g$ is $L_g$-Lipschitz, then*

$$\text{disc}_{\ell_{CE}}(f\#\widehat{P}_w, f\#Q) \leq L_f L_g W_1(f\#\widehat{P}_w, f\#Q).$$

*Thus,*

$$R_Q(h) \leq R_{\widehat{P}_w}(h) + L_f L_g W_1(f\#\widehat{P}_w, f\#Q) + \lambda^\star + \mathcal{E}_n.$$

*Since anchor alignment contracts $L_f$ (Lemma 3.1), this bound tightens.*

**Corollary 3.2** (KL/Rényi control via exponential tilting). *Let $w(x, y) \propto \exp\{\tau s(x, y)\}$ with bounded score $s$ (cosine alignment). If $\ell_{CE} \in [0, 1]$ is $\sigma^2$-sub-Gaussian under $\widehat{P}_w$, then*

$$\mathbb{E}_Q \ell_{CE} \leq \mathbb{E}_{\widehat{P}_w} \ell_{CE} + \sigma\sqrt{2\,\text{KL}(Q\|\widehat{P}_w)}.$$

*Equivalently, using Rényi divergence of order $\alpha > 1$,*

$$R_Q(h) \leq \exp\Big(\tfrac{\alpha-1}{\alpha} D_\alpha(Q\|\widehat{P}_w)\Big) \Big(\mathbb{E}_{\widehat{P}_w} \ell_{CE}^{\frac{\alpha}{\alpha-1}}\Big)^{\frac{\alpha-1}{\alpha}}.$$

Taken together, Theorem 3.2 and its corollaries show that $A^3W$ reduces the generalization gap by acting directly on the discrepancy term that appears in domain generalization bounds. Anchor alignment contracts the Lipschitz constant of the representation (Corollary 3.2), while softmax-based reweighting moves the empirical distribution $\widehat{P}_w$ closer to the clean source $P^\star$ and reduces $\mathrm{KL}(Q\|\widehat{P}_w)$ (Corollary 3.3). This theoretical view suggests that beyond classification accuracy, $A^3W$ should also lower empirical discrepancy measures between training and held-out domains. In the experiment section below, we confirm this prediction empirically by showing that $A^3W$ indeed reduces source–target discrepancy and yields consistent accuracy gains under both domain shifts and label noise.

Table 1: Dataset Information for Domain Generalization Benchmarks

| Dataset | Domains | # Classes | Class Descriptions | # Images |
|---|---|---|---|---|
| PACS | Photo, Art, Cartoon, Sketch | 7 | Dog, Elephant, Giraffe, Guitar, Horse, House, Person | 9,991 |
| VLCS | Caltech101, LabelMe, SUN09, VOC2007 | 5 | Bird, Car, Chair, Dog, Person | 10,729 |
| OfficeHome | Art, Clipart, Product, Real | 65 | Office/home objects | 15,500 |
| SVIRO | 4 Car Makes | 7 | Description of back seat | 25,000 |
| DomainNet | Clipart, Infograph, Painting, Quickdraw, Real, Sketch | 15 | Common Objects | 25,730 |

## 4 Experiments

### 4.1 Experimental Setup and Datasets Preprocessing

We conducted our experiments on a server with 10 NVIDIA Quadro RTX 6000 24G GPUs. We extensively evaluate our method on domain generalization with noisy data using four benchmark datasets. For more details, please refer to their original publications. **PACS** (Li et al., 2017) is a 7-class image classification dataset that spans four distinct domains (Photo, Art Painting, Cartoon, and Sketch). Renowned for its diverse artistic styles, PACS offers a challenging testbed for robust representation learning. **VLCS** (Fang et al., 2015) comprises 5 classes drawn from four domains (Caltech101, LabelMe, SUN09, and VOC2007). Each domain originates from a different source, resulting in significant variability in image characteristics and presenting a broad generalization challenge with both natural and scene-centric images. **Office-Home** (Venkateswara et al., 2017) is a 65-class dataset designed to capture common objects in everyday office and home environments. Its wide range of object categories and the substantial variation in style and background make it a rigorous benchmark for domain generalization. **SVIRO** (Cruz et al., 2020) is a synthetic dataset focused on vehicle interiors. It contains 25,000 images from 10 distinct vehicle interior environments and features 7 occupant classes. SVIRO provides a challenging scenario for domain generalization, especially in handling variations in interior design and occupancy. **DomainNet** (Peng et al., 2019) is a large-scale dataset spanning six domains (Clipart, Infograph, Painting, Quickdraw, Real, Sketch) with 345 classes. It encompasses a broad range of styles and object categories, capturing significant distribution shifts across these domains. Due to computation resource limitation, we only used the first four domains in SVIRO and 15 classes per domain for DomainNet. More information is provided in Table 1.

All experiments are conducted using DomainBed (Gulrajani & Lopez-Paz, 2020) to ensure consistency across datasets. To assess the out-of-distribution performance of various algorithms, we inject instance-independent symmetric label noise based on (Qiao & Low, 2024) where we add 10% and 25% label noise. For all presented real-world datasets, we employ a ResNet-50 (He et al., 2016) pretrained on ImageNet (Deng et al., 2009), and standard data augmentation techniques are applied across all experiments. For the domain-shift datasets, we used the same metric as (Qiao & Low, 2024), in which we perform single-domain cross-test experiments by designating each domain in turn as the test set and using the remaining domains for training. In these experiments, we use 20% of the test data for model selection, and no early stopping is applied. For each dataset, we run 3 independent trials; in each trial, we perform a hyperparameter search over 20 different configurations according to DomainBed's default settings. Each environment were made to be the target dataset independently, thus 240 total trials were performed for each dataset. Specifically, we train all models for 5000 steps on all real-world datasets.

Table 2: Cross-test accuracy (%) for domain shifts under a noise level of $\eta = 0.25$. Best results in **bold**.

| Method | PACS | VLCS | Office-Home | SVIRO | DomainNet |
|---|---|---|---|---|---|
| ERM | $74.5 \pm 0.6$ | $71.9 \pm 0.6$ | $54.9 \pm 0.3$ | $77.9 \pm 0.9$ | $65.0 \pm 0.4$ |
| GroupDRO | $74.7 \pm 0.4$ | $71.2 \pm 0.2$ | $54.1 \pm 0.3$ | $75.4 \pm 0.5$ | $68.4 \pm 0.2$ |
| IRM | $71.0 \pm 1.9$ | $70.3 \pm 0.3$ | $53.9 \pm 1.8$ | $62.7 \pm 14.7$ | $37.6 \pm 15.5$ |
| VREx | $73.5 \pm 0.7$ | $71.8 \pm 0.6$ | $53.0 \pm 0.9$ | $73.6 \pm 2.6$ | $64.9 \pm 0.8$ |
| Mixup | $75.2 \pm 0.9$ | $71.9 \pm 0.5$ | $57.5 \pm 0.3$ | $83.3 \pm 1.5$ | $67.9 \pm 0.2$ |
| URM | $72.1 \pm 0.3$ | $70.2 \pm 0.9$ | $58.1 \pm 0.3$ | $83.6 \pm 1.5$ | $66.1 \pm 1.9$ |
| $A^3W$ (ours) | $\mathbf{82.1 \pm 0.5}$ | $\mathbf{76.1 \pm 0.3}$ | $\mathbf{65.2 \pm 0.2}$ | $\mathbf{93.9 \pm 0.1}$ | $\mathbf{73.9 \pm 0.2}$ |

### 4.2 Baseline Methods

We compare $A^3W$ against a suite of representative **baseline methods** from the domain generalization literature (Qiao & Low, 2024):

- *ERM*: Empirical Risk Minimization, which trains a single model on the aggregated source domains (Vapnik, 1991).

- *Mixup*: Mixes pairs of source samples to create interpolated training examples, improving robustness (Zhang et al., 2017).

- *GroupDRO*: Optimizes worst-group loss among source domains to handle distribution shifts (Sagawa et al., 2019).

- *IRM*: Invariant Risk Minimization, which enforces domain-invariant representations to improve out-of-distribution generalization (Arjovsky et al., 2019).

- *V-REx*: Variance Risk Extrapolation, which penalizes variance in risk across different environments to improve generalization (Krueger et al., 2021).

- *URM*: Uniform Risk Minimization, which encourages uniformly distributed final-layer feature representations to improve OOD robustness and group fairness without using group/domain labels (Krishnamachari et al., 2024).

We obtain the comparison results from Qiao & Low (2024), and we use the Pytorch suite developed by Gulrajani & Lopez-Paz (2020) for base for algorithm implementation. The subset of TerraIncognita from Qiao & Low (2024) was no longer publicly available, so we used the SVIRO dataset. The SVIRO dataset's baseline method result was obtained by running the experiment in the same setup as Qiao & Low (2024). The target domain is divided into validation and test sets. The validation set is used to tune the hyperparameters by selecting the configuration that yields the highest accuracy. Using the optimal hyperparameter configuration, we then evaluate the model's performance on the test set, and the average classification accuracy on the target domain is used as the evaluation metric.

### 4.3 Experimental Results

Table 2 and 3 present the average test accuracy for different values of $\eta = 0.1$ and $\eta = 0.25$, respectively. The experimental results show that $A^3W$ consistently outperforms all baseline methods. We observe the following:

1) $A^3W$ consistently achieves the highest accuracy across all domains in all datasets. Its margin of improvement often exceeds 6–8% on average compared to the baseline methods, indicating stable training dynamics, leading to consistent accuracy improvements across multiple datasets.

Table 3: Cross-test accuracy (%) for domain shifts under a noise level of $\eta = 0.1$. Best results in **bold**.

| Method | PACS | VLCS | Office-Home | SVIRO | DomainNet |
|---|---|---|---|---|---|
| ERM | $82.0 \pm 0.5$ | $75.0 \pm 0.3$ | $62.2 \pm 0.1$ | $80.8 \pm 1.9$ | $70.8 \pm 0.5$ |
| GroupDRO | $82.4 \pm 0.3$ | $75.1 \pm 0.1$ | $61.3 \pm 0.4$ | $78.6 \pm 2.1$ | $71.9 \pm 0.1$ |
| IRM | $80.4 \pm 1.2$ | $74.6 \pm 0.3$ | $61.2 \pm 1.2$ | $72.5 \pm 8.1$ | $38.9 \pm 19.8$ |
| VREx | $81.4 \pm 0.2$ | $75.0 \pm 0.1$ | $60.6 \pm 0.5$ | $81.6 \pm 3.5$ | $66.9 \pm 0.4$ |
| Mixup | $83.6 \pm 0.1$ | $75.5 \pm 0.2$ | $63.9 \pm 0.1$ | $84.1 \pm 0.7$ | $69.2 \pm 0.6$ |
| URM | $80.1 \pm 0.4$ | $75.3 \pm 0.2$ | $63.8 \pm 0.2$ | $79.2 \pm 0.7$ | $71.9 \pm 0.4$ |
| $A^3W$ (ours) | $\mathbf{85.2 \pm 0.2}$ | $\mathbf{78.5 \pm 0.2}$ | $\mathbf{68.2 \pm 0.3}$ | $\mathbf{94.8 \pm 0.3}$ | $\mathbf{76.1 \pm 0.1}$ |

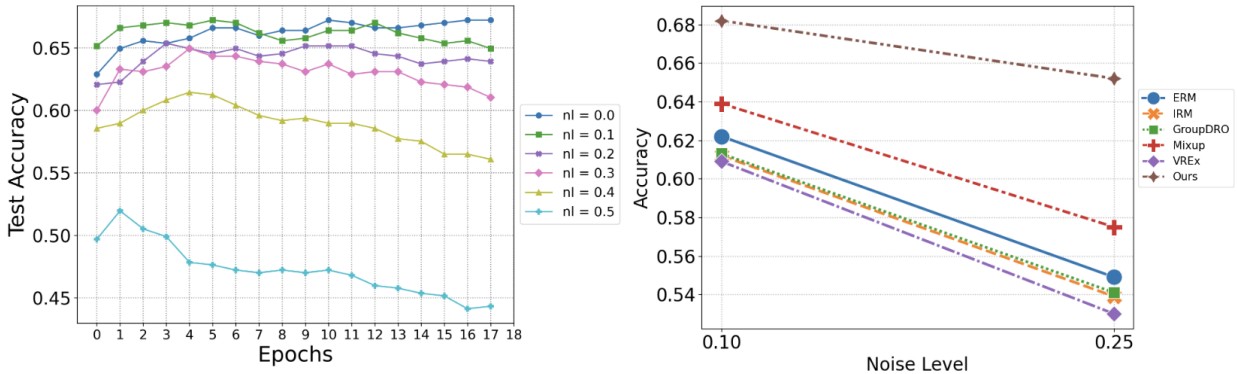

Figure 3: Noise analysis. (a) The effect of increasing noise levels on classification accuracy reveals that higher noise leads to a sharper decline, reflecting an increased tendency to overfit. (b) $A^3W$ is most robust to noise injection, with its accuracy decreasing by only 0.2 when noise increases from 0.1 to 0.25, in contrast to other algorithms, which show declines between 0.427 and 0.527.

2) $A^3W$ shows the strongest performance gain in SVIRO (averaged to 13%), where the most semantic information is given, proving the importance of the NLP anchor in guiding the sampling process of learning the featurizer.

3) Among the baselines, **Mixup** frequently attains the second-best performance, suggesting that interpolation-based augmentation can help mitigate moderate noise. However, its improvements are relatively small and are inconsistent across different domains.

4) In our noise analysis (see Figure 3), we observe that the general training accuracy tends to drop earlier as noise levels increase, a sign of overfitting to noise. Among all evaluated methods, $A^3W$ exhibits the smallest decline in accuracy with rising noise, demonstrating its resilience to overfitting under noisy conditions. Additionally, the flexibility of simple loss reweighting in handling noisy labels allows it to adapt well to various network architectures. Thus, in subsequent experiments, we use $A^3W$ as the default approach unless otherwise specified.

## 4.4 Ablation Study

To better understand the role of each component in $A^3W$, we conduct ablation studies under several key questions, as summarized in Table 4. The experiments span multiple datasets, and the results highlight how removing NLP anchor alignment, omitting the softmax weighting mechanism, or perturbing the semantic structure of anchors impacts overall performance.

Table 4: Cross-test accuracy (%) for domain shifts. Best results in **bold**.

| Method | PACS | VLCS | Office-Home | SVIRO | DomainNet |
|---|---|---|---|---|---|
| w/out NLP anchor | 80.7 | 75.5 | 65.1 | 91.7 | 85.5 |
| w/out weighted loss | 79.7 | 74.2 | 64.7 | 93.6 | 85.8 |
| Random anchors | 73.5 | 47.5 | 57.0 | 60.4 | 60.5 |
| Shuffled anchors | 77.2 | 65.6 | 55.0 | 61.5 | 70.7 |
| $A^3W$ (baseline) | **82.1** | **76.1** | **65.2** | **93.9** | **86.5** |

*1) Why does removing NLP anchor alignment reduce performance?* In Table 4, discarding the NLP anchor alignment step (i.e., "w/out NLP anchor") yields an average accuracy drop of about 0.86% across the listed datasets. This decrease arises because anchor alignment provides semantic guidance that helps the model differentiate meaningful features from spurious correlations. Without alignment, the feature extractor lacks an external semantic reference, making it more prone to overfitting on noisy labels and domain-specific artifacts.

*2) How does eliminating softmax weights affect training stability?* When we replace the adaptive softmax weighting with uniform weights (i.e., "w/out weighted loss"), the accuracy declines by an average of 1.16% across the datasets. This finding suggests that the continuous weighting scheme (Section 3.8) is crucial for emphasizing well-aligned (likely clean) samples. In contrast, uniform weighting fails to down-weight noisy or misaligned samples, reducing training stability and overall performance.

*3) What happens if semantic anchors are replaced with random or shuffled vectors?* To further validate the importance of CLIP-derived semantic structure, we introduce two additional ablations: *Random anchors*, where each class is assigned a Gaussian random vector of the same dimensionality as CLIP text embeddings; and *Shuffled anchors*, where CLIP embeddings are computed normally but randomly permuted across classes. Both settings remove the intended semantic correspondence between labels and anchor vectors, while preserving the optimization procedure. As shown in Table 4, both variants significantly degrade performance across all datasets, with random anchors being especially harmful (e.g., 73.5% vs. 82.1% on PACS). Notably, the model still achieves non-trivial accuracy in these cases. This is because the backbone network and classifier can still learn discriminative features from the raw input, and the mapping layers continue to receive gradient signals even if the anchors themselves are meaningless. However, without semantically consistent anchors, the learned representation lacks a reliable external reference, leading to poorer alignment across domains and ultimately weaker generalization.

*4) Are alignment, weighting, and semantic structure jointly necessary for robust generalization?* Finally, the baseline $A^3W$ model that integrates NLP anchor alignment, semantic consistency, and softmax weighting achieves the highest accuracy on average (e.g., 82.1% on PACS). These results indicate that semantic anchors and adaptive weighting reinforce each other: alignment ensures meaningful feature extraction, while the weighting mechanism selectively highlights reliable data. Removing either component, or corrupting the anchor structure, leads to noticeable performance degradation, underscoring their combined importance for robust domain generalization under noisy labels.

These findings confirm that semantic alignment with textual anchors and the associated softmax weighting plays a pivotal role in boosting generalization. The additional MA network contributes incremental stability but is not solely responsible for the performance gains.

### 4.5 Convergence Analysis and Feature Clustering

In addition to the experimental results, we further assess the feasibility of our approach by examining its trainability and convergence properties. Figure 4 illustrates the convergence behavior of our model as the training step size increases. The convergence trajectories, shown under three different random seeds

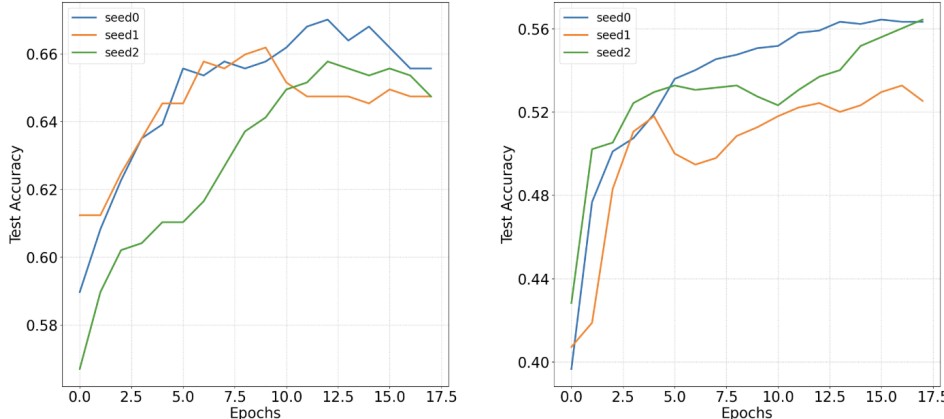

Figure 4: Convergence trajectories of A3W compared with ERM and Mixup under three random seeds on PACS and VLCS. The x-axis denotes training steps, and the y-axis denotes average accuracy (%).

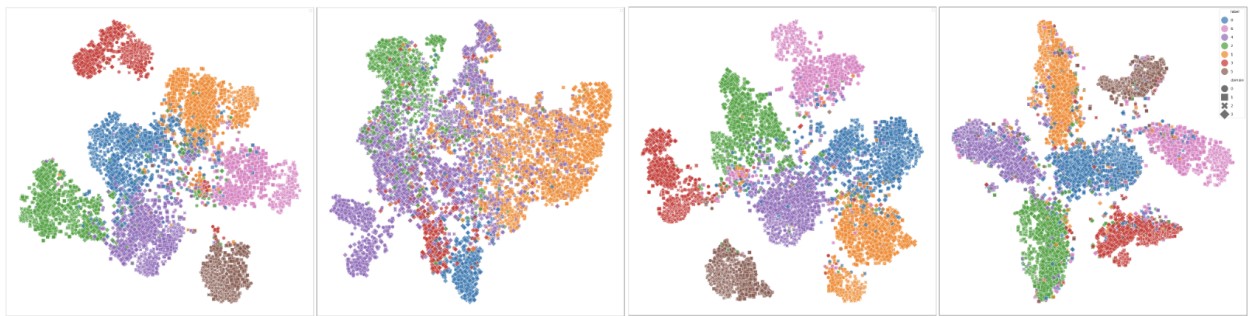

Figure 5: t-SNE embeddings on PACS. Each point represents one sample; color indicates class and marker shape indicates domain. From left to right are ERM, IRM, MixUp, and $A^3W$. Compared to ERM, IRM, and Mixup, $A^3W$ yields more compact and well-separated clusters.

and training configurations across two datasets, demonstrate that $A^3W$ achieves stable optimization and effectively minimizes the overall loss, even in the presence of label noise. This stability validates our method's robustness and its ability to learn domain-invariant representations. Furthermore, Figure 5 presents t-SNE embeddings of features learned by three different methods on the PACS dataset. In these visualizations, each point represents a sample, with color indicating the class and marker shape indicating the domain. By comparing the best models from each approach, it is evident that the proposed $A^3W$ produces more cohesive and well-separated clusters across both training and test domains, underscoring its superior capability in learning discriminative features.

## 4.6 Parameter Sensitivity

We consider four key hyperparameters in our method: the regularization parameter ($\lambda$), the iteration frequency, the temperature ($\tau$), and the learning rate (lr). For fairness, we keep all other settings fixed while varying one parameter at a time, and we set the random seed for each trial. Figure 6 presents the parameter sensitivity analysis, with red markers indicating the optimal performance points. The best performance is achieved at $\lambda = 0.1$, though the accuracy remains competitive even when $\lambda$ deviates slightly from this value, demonstrating resilience to parameter shifts. Similarly, increasing the iteration frequency from 5 to 20 results in only minor changes in accuracy, which underscores the method's stability. For the temperature, varying $\tau$ from 5 to 15 shows that the optimal performance is attained at $\tau = 10$, with a sharp decline in accuracy observed when $\tau$ is raised to 15. This suggests that an excessive temperature causes the amplification in the softmax to become overly aggressive, rendering it too selective and potentially discarding useful information; however, when properly tuned, it also helps to effectively diminish the effect of samples that deviate from

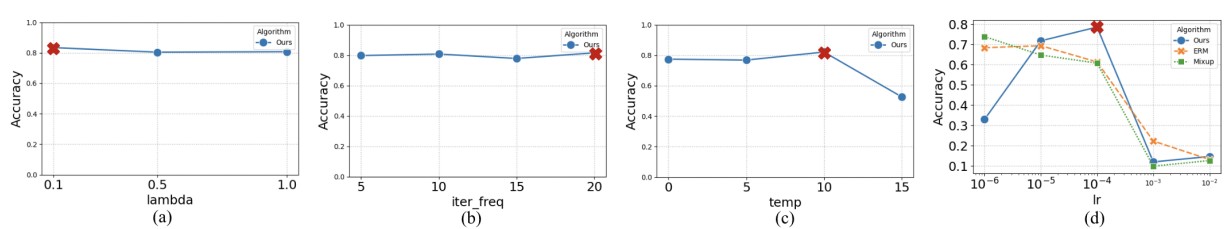

Figure 6: Parameter sensitivity analysis. Parameter sensitivity analysis of $A^3W$. Each subplot varies one hyperparameter while fixing others: (a) $\lambda$ (regularization weight), (b) update frequency, (c) $\tau$ (temperature), and (d) learning rate. The x-axes denote hyperparameter values and the y-axes denote accuracy (%). Red markers indicate the optimal settings.

the semantic anchors. Although Theorem 3.1 guarantees that a sufficiently high temperature will favor uncorrupted data through the softmax weighting, the key is finding the right balance–high enough to filter out noise without overshooting. $A^3W$ achieves the highest accuracy at $10^{-4}$, as indicated by the red cross mark, but performance drops significantly at higher learning rates. Both ERM and Mixup exhibit more stable performance across lower learning rates but also experience a sharp decline at $10^{-3}$ and beyond. Overall, these findings demonstrate that our approach exhibits robustness across a broad spectrum of hyperparameter settings, maintaining strong performance even under suboptimal conditions. Furthermore, they suggest that careful hyperparameter tuning can further improve performance, offering a competitive advantage over alternative methods.

## 4.7 Impact of Semantic Richness in NLP Anchors

Our ablation studies reveal that NLP anchors provide effective guidance, a finding further supported by experimental results on the SVIRO dataset (see Table 2 and Table 3). In particular, when descriptive class names such as "car back seat," "infant car seat," and "child in convertible car seat" were used, $A^3W$ achieved the highest performance improvement among all datasets. In contrast, other datasets used single-word labels showed less pronounced gains. However, this reliance on rich semantic information may pose challenges when the categories are very similar or less frequently encountered. We ruled out the possibility that the number of classes was a limiting factor—our experiments on the OfficeHome dataset, which includes 65 classes, still demonstrated performance gains over the baselines. Future work could explore more tailored network architectures for instance-specific domain generalization, especially in scenarios involving closely related classes.

## 4.8 Training Noise as an Implicit Regularizer

In some parameter settings, we observed that introducing more noise during training can actually lead to improved accuracy, as shown in Fig. 3. This counterintuitive result occurs because additional noise acts as a form of regularization. Essentially, it prevents the model from overfitting to spurious correlations and irrelevant details in the training data. Previous work (Chen et al., 2023) suggests that this injected noise encourages the network to learn more robust and invariant features by smoothing the loss landscape and promoting the discovery of flatter minima. As a result, even though the training data is noisier, the model is better able to capture the underlying patterns that generalize well to the target domain. This leads to improved performance on unseen data, as the network becomes less sensitive to the peculiarities of the noisy training samples.

## 4.9 Experiments under Asymmetric Noise

While our primary experiments focus on symmetric label noise for consistency with prior work, real-world datasets often exhibit more structured corruption. A common case is *asymmetric noise*, where labels are flipped to semantically confusable classes (e.g., "dog" → "horse"). We evaluate the robustness of $A^3W$ under this setting by conducting additional experiments on PACS with asymmetric noise applied at three corruption levels ($\eta \in \{0.1, 0.2, 0.3\}$). Here, $\eta$ denotes the probability that a training sample's label is flipped

to a pre-specified confusable class according to the asymmetric transition matrix. For instance, $\eta = 0.1$ corresponds to 10% of labels being corrupted, while the remaining 90% remain clean.

Since Mixup consistently achieved the strongest performance among existing domain generalization baselines in the symmetric-noise setting, we restrict our comparison here to Mixup and $A^3W$. Table 5 summarizes the results. While Mixup's performance drops sharply as the noise level increases, $A^3W$ maintains high accuracy across all settings, with only minor degradation even at $\eta = 0.3$. These findings confirm that the robustness of $A^3W$ extends beyond symmetric corruption and holds under more realistic asymmetric noise processes.

Table 5: Cross-test accuracy (%) on PACS under asymmetric label noise at different noise rates. Best results in **bold**.

| Method | $\eta = 0.1$ | $\eta = 0.2$ | $\eta = 0.3$ |
|--------|--------------|--------------|--------------|
| Mixup  | $94.8 \pm 1.2$ | $80.0 \pm 0.6$ | $78.7 \pm 0.1$ |
| **A3W** | $\mathbf{96.5 \pm 0.3}$ | $\mathbf{93.3 \pm 0.3}$ | $\mathbf{90.3 \pm 1.1}$ |

### 4.10 Cosine Similarity Loss for Feature Representation Learning

Negative cosine similarity is commonly used as a loss function when learning feature representations, especially in tasks such as metric learning, similarity-based learning, and contrastive learning. We compared L2 and cosine similarity, where cosine similarity greatly outperformed L2 loss. Although both encourage the learning of representations that are close together for similar items (i.e. samples from the same class) and far apart for dissimilar ones to support the formation of a more discriminative and robust feature space, cosine similarity captures the orientation of the vectors rather than their magnitude. In other words, cosine similarity is scale invariant: if a vector is scaled by a constant factor, its cosine similarity with another vector remains unchanged, whereas L2 loss is directly affected by changes in magnitude. This property is particularly advantageous in tasks where the relative direction of feature vectors (which captures semantic information) is more important than their absolute values. Additionally, cosine similarity naturally emphasizes the alignment between vectors, encouraging features from the same class to be directionally similar, regardless of their scale. This can lead to more robust clustering and improved generalization, especially when the features may vary in magnitude due to factors unrelated to the underlying semantics.

## 5 Discussion

Despite these promising results, $A^3W$ has several limitations that warrant discussion. One important aspect is computational cost. Compared to baseline DG methods such as ERM or CORAL, our approach introduces additional mapping layers for each class and computes cosine similarity loss with softmax-based weighting, which increases both parameter count and training time. While our experiments indicate that the overhead is manageable on current benchmarks (approximately a 10% increase in runtime), scaling $A^3W$ to large-scale datasets with hundreds or thousands of classes (e.g., ImageNet) may render the current formulation less practical. We recognize this scalability concern and outline several promising directions to address it. First, anchor sharing could allow semantically related classes to reuse projection heads (e.g., "tiger" and "lion"), thereby reducing redundancy while preserving discriminability. Second, class clustering of NLP anchors into higher-level semantic groups would enable shared projections across clusters, amortizing computational cost while retaining semantic alignment. Third, low-rank factorization of projection matrices could cut parameter count in high-dimensional embedding spaces such as those produced by CLIP. Finally, hierarchical anchoring provides a coarse-to-fine mechanism (e.g., "animal → mammal → dog"), ensuring that only a subset of anchors is active for each sample. These strategies represent natural extensions of $A^3W$ that would preserve the benefits of semantic grounding while improving scalability.

Beyond computational scalability, there are additional opportunities to enhance $A^3W$. A promising direction involves adaptive anchor selection. Currently, $A^3W$ assumes a fixed set of text embeddings throughout training, which may not always be optimal. Instead of static anchors, one could explore dynamic anchor refinement, where text embeddings evolve based on learned feature distributions. This strategy would enable

the model to refine its representations over time, thereby improving alignment with the most semantically relevant concepts. Additionally, integrating contrastive learning techniques could further strengthen the alignment between visual and textual representations. Future work could also extend $A^3W$ to multi-label and hierarchical classification settings. Many real-world applications involve objects that belong to multiple categories simultaneously (e.g., an image of a wolf could be categorized as both "canine" and "wild animal"). Current $A^3W$ anchors operate at the class level, but incorporating hierarchical category embeddings (e.g., WordNet-based representations) could improve generalization by capturing higher-level semantic relationships. Similarly, expanding the algorithm to support multi-modal domain generalization or incorporating additional cues such as audio or structured metadata could further enhance robustness.

## 6 Conclusion

In this paper, we introduced $A^3W$, a simple yet effective domain generalization algorithm under noisy data that integrates external knowledge from large-scale language models to guide visual feature learning. By mapping learned representations to class-specific NLP anchors, we impose additional semantic constraints that mitigate domain-specific biases. Experimental results across five benchmark datasets demonstrate consistent improvements over state-of-the-art methods, underscoring the effectiveness of knowledge-guided strategies in dealing with unseen domains to the best of our knowledge.

Moreover, our ablation studies reveal that both semantic alignment and adaptive weighting play pivotal roles in enhancing the robustness of our approach. Notably, our algorithm's modular design allows for easy replacement or customization of its individual components (e.g. integrating different moving average strategies or alternative weighting schemes) to better suit diverse tasks. This flexibility not only simplifies the process of adapting to new challenges but also sets the stage for future explorations in knowledge-guided domain generalization. As vision-language models continue to evolve, the potential to further refine and extend these methods promises even greater strides toward interpretable, robust, and versatile deep learning systems.

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
