# OpenReview forum: "A Language Anchor-Guided Method for Robust Noisy Domain Generalization"
_TMLR — Rejected by TMLR_

### Review · Reviewer_iJEw · 2025-06-26

**Summary Of Contributions:**

The authors propose a novel method to address two key challenges in domain generalization: distribution shift and label noise. Their approach leverages label text embeddings as anchors, assigning sample weights based on the distance between samples and their corresponding anchors. Experimental results demonstrate significant performance improvements over existing domain generalization techniques.

**Audience:**

Yes

**Claims And Evidence:**

No

**Requested Changes:**

1. While the manuscript includes an assumption about label noise and offers theoretical insights, it is important to clearly state the final guarantees derived from the theory. Specifically, the impact of label noise on these guarantees should be explicitly discussed and quantified, if possible.

2. The current anchor is based solely on a plain textual description of the label. It would be more compelling to consider whether incorporating style-aware textual descriptions, reflecting different domains or image styles—alongside the label could lead to more representative and effective anchors.

3. An ablation study evaluating the contribution of the anchor embedding to the overall performance is strongly recommended to validate its effectiveness.

4. The experimental results for baseline methods are taken directly from another paper. It is unclear whether the evaluations were conducted under the same data conditions, especially given the stochastic nature of label noise injection. Additional clarification and reproducibility assurance are needed.

**Strengths And Weaknesses:**

Strengths

1. The architecture of the proposed method, as illustrated in Figure 2, is clear and easy to follow.

2. The paper addresses a problem of practical relevance in domain generalization.

3. The introduction of label embeddings as anchors to mitigate domain drift is an interesting and intuitive idea.


Weaknesses

1. The proposed architecture does not fully align with the stated motivation. Since the anchor is derived solely from the label, it remains identical across domains for the same class. This overlooks potentially useful information contained in the domain-specific style or context. A richer anchor representation incorporating both label and style descriptions may better reflect the intended objective.

2. The benefits of the architecture in handling label noise are not theoretically justified.

3. While the paper discusses label noise and provides some theoretical insights, it lacks guarantees regarding gradient stability (e.g., whether it is bounded by a parameter $\gamma$). Moreover, the role of the label noise rate parameter $p$ in determining final performance is not sufficiently analyzed.

4. The experimental results for baselines are taken from a prior work [Qiao & Low, 2024], yet the method proposed in that work is not included as a direct baseline for comparison. Furthermore, the procedure for injecting label noise may introduce randomness—particularly in imbalanced datasets—raising concerns about reproducibility and whether all methods were evaluated under consistent conditions.

---

> ### Author Response · Authors · 2025-09-20
>
> We appreciate the reviewer’s insightful comments, and we address them below.
>
> **Richer anchor representation (W1 & RC2) :** As noted in our Discussion section, our current design derives anchors solely from class labels, which makes them domain-invariant by construction. While this approach does overlook potentially useful style or context information, it allows us to better isolate the effect of noisy labels on domain generalization. Extending the anchor representation to incorporate domain-aware prompts is a valuable future direction; prior work such as DiPrompt (Bai et al., 2024) highlights the promise of disentangled, domain-specific prompt tuning. We have also experimented with adding domain names into the text prompts, but these modifications did not lead to consistent improvements. This outcome suggests that, in our current setup, the primary benefit of anchors lies in their domain-invariant nature. Because the anchor representation remains identical across domains for a given class, it captures semantic class identity rather than style or context specific factors, thereby offering a stable reference that mitigates the influence of label noise.
>
> **Theoretical justification (W2, RC1):** Our focus in this work is on understanding the interplay between label noise and domain generalization. The theoretical impact of label noise has been carefully studied in Qiao et al. (2024), and our contribution complements these results by emphasizing empirical consequences. Figure 1 illustrates how noise impairs domain-invariant learning, and Figure 3 shows the corresponding effect on classification accuracy. For added clarity, we will provide a table quantifying performance at each noise level in the final version. While our emphasis lies primarily in empirical validation, we provide theoretical insights in Section 3.8, where we explain how the semantic anchors offer a soft regularization mechanism that mitigates noisy labels’ influence. Gradient behavior under label noise is further supported by the convergence trajectory shown in Figure 4, which exhibits greater stability compared to baselines. Figure 3 illustrates that our method exhibits the least performance degradation as noise increases, underscoring its robustness.
> Furthermore, Section 3.8 has been expanded with a formal generalization bound and corollaries under Wasserstein and KL/Rényi instantiations, linking A3W’s anchor alignment and softmax-based reweighting to reduced discrepancy terms. Together, these theoretical and empirical results clarify why our method achieves robustness improvements under both domain shifts and noisy labels.
>
> **Reproducibility Concern (W4, RC4)**: Thank you for highlighting this point. We use results from Qiao & Low (2024) primarily as a representative benchmark, as their work provides a comprehensive theoretical and empirical study of current domain generalization algorithms under label noise. Their method is analytical in nature and not intended as a standalone algorithm for comparison. To ensure fair comparison and mitigate variability, we strictly followed the reproducibility protocols provided in Qiao & Low (2024), including their data splits and noise injection procedures. As their work focuses on benchmarking rather than proposing a new method, we treated it as a reference point rather than a baseline competitor.
>
> **Additional Ablations (W3, RC3)**: We address this through the ablation study presented in Table 4. Specifically, the first row shows performance without the NLP anchor, where we observe a consistent accuracy drop across all datasets. This confirms that the semantic anchor meaningfully contributes to feature guidance and model robustness. In addition, we added two further ablation studies (random anchors and shuffled anchors) to strengthen the evidence for the effectiveness of our approach. Both settings remove the intended semantic correspondence between labels and anchor vectors, while preserving the optimization procedure. Both variants significantly degrade performance across all datasets, with random anchors being especially harmful (e.g., 73.5% vs. 82.1% on PACS).
>
> We include these details in the updated version. In addition, we are committed to open-sourcing the code upon acceptance, and we will promptly incorporate any further feedback.

---

### Review · Reviewer_oqsS · 2025-07-12

**Summary Of Contributions:**

The authors introduce A3W, a method to improve domain generalization, particularly focusing on increasing the robustness to noisy labels. A3W leverages CLIP embeddings as class-wise anchors to guide both feature alignment and adaptive sample weighting. The method is evaluated against five standard DG baselines across five multi-domain image classification datasets. The authors provide theoretical proof and empirical evidence demonstrating that A3W improves both the feature extractor and "filters out" noisy samples from the loss computation.

**Audience:**

Yes

**Claims And Evidence:**

No

**Requested Changes:**

1.  Correct the clarity issues mentioned. Additionally, the overall presentation could be improved. The paper contains a large amount of text but lacks comprehensive visual illustrations, which makes it harder to follow and digest the key ideas.
2. Include newer DG methods in Tables 2 and 3 to strengthen your claim for SOTA. Including at least other contrastive-based DG methods like PCL, which are close contemporaries in terms of idea and usage, would help.
4. Please clarify the exact claims, motivations, and conclusions you reach in the paper. (wrt to weakness 1, 2, and 3).

**Strengths And Weaknesses:**

**Strengths:**

1. The authors claim A3W consistently achieves 6–8% of accuracy improvement on average compared to the baseline methods.
2. The study is extensive in terms of the number and diversity of datasets used.


**Weakness:**

1. The claim "We introduce the concept of NLP anchors derived from large-scale language models (e.g., CLIP),
which provide domain-invariant and semantically rich feature constraints that significantly improve.
Model robustness. " feels exaggerated, as CLIP embeddings have been used for improving domain generalization in  "A Sentence Speaks a Thousand Images: Domain Generalization through Distilling CLIP with Language Guidance," ICCV 2023.
2.  Similarly, the idea of class-wise references (anchors or proxies) has been previously used in PCL: Proxy-Based Contrastive Learning for Domain Generalization, CVPR.
3. In Section 4, we see that out of the claimed 6% performance increase, removing NLP anchor alignment results in a 0.86% drop in average accuracy, and removing adaptive weighting results in a 1.16% drop. This raises the question: Where does the rest of the improvement come from? What would happen if we remove both (freeze the feature extractor, learn only the projection layer, and use uniform weighting )?
4. The baselines used are limited to methods released before or around 2021: ERM, Mixup, IRM, GroupDRO, VREx. Newer DG methods are not included, which weakens the comparative claim of “state-of-the-art” performance.
5. A lot of clarity issues make it hard to follow the paper:
- Figure 1(d): “Noise accuracy” is never clearly defined
- Equation 14 notation is unclear.
- Figures 4–6: Labels are minimal and not self-contained.

---

> ### Author Response · Authors · 2025-09-20
>
> We thank the reviewer for their valuable feedbacks. Below are our replies:
>
> **Difference with Prior Work (W1, W2):** Comparing two previous papers, we summarize our differences with them as follows. Overall, our paper proposes a different objective and problem setting. In RISE, the objective is to transfer semantic invariance from CLIP’s text encoder into a smaller student model with absolute and relative distance losses. In PCL, the objective is to improve generalization by replacing sample-to-sample contrastive pairs with learned proxies. However, in our method, we specifically designed it to address both domain shift and label noise, where neither previously mentioned paper included any mechanism for label-noise mitigation. If we summarize our approach in several aspects, the difference is as follow:
> 1. Semantic reference and representation alignment: RISE uses CLIP for generating text embeddings, then aligns the image using absolute and relative distance losses. PCL uses learned class in the visual space, without semantic information. Our method uses CLIP text encoder to generate fixed class anchors, then projects image features into semantic space via cosine similarity.
> 2. Weighting mechanism and noise robustness: RISE uses no weighting, and only uses CE with distance loss. CPL also uses no weighting, and uses standard softmax on proxy similarity. Both methods assumes clean labels and are not designed for handling noise. Our method proposes adaptive weighting via softmax with NLP anchor to down-weight noisy labels.
> 3. In Section 3.8, we formalize how anchor alignment acts as a regularizer (Lemma 3.1) and prove that our adaptive weighting mitigates the effect of noisy labels (Theorem 3.1). We further provide a generalization bound specialized to our method (Theorem 3.2) with Wasserstein and KL/Rényi instantiations. These results show that alignment contracts the representation space, while reweighting shifts the empirical distribution closer to the clean one, thereby reducing discrepancy.
> In short, while RISE and PCL provide valuable guidance and share related ideas, our contribution lies in adapting these principles to a new setting that jointly considers domain generalization and label noise, and in demonstrating both theoretically and empirically how this adaptation yields improved robustness. We have now cited both papers appropriately in our manuscript and framed them as complementary prior work.
>
> **Ablation Clarification (W3, RC1):** The concern about where the remainder of the reported 6% gain comes from is largely addressed by the first ablation: once NLP anchor alignment is removed, the feature extractor is effectively frozen and the model relies only on the projection layer with uniform weighting, so this already captures the “remove both” setting. To further strengthen the analysis, we added ablations with random anchors and shuffled anchors. Both settings keep the training pipeline unchanged but break the semantic structure of the anchors. The results show sharp drops: for example, on VLCS accuracy falls from 76.1% (ours) to 47.5% with random anchors and 65.6% with shuffled anchors, and on DomainNet it drops from 86.5% to 60.5% and 70.7%, respectively. Similar though smaller gaps appear on PACS, Office-Home, and SVIRO. These numbers demonstrate that the observed gains do not come from auxiliary vectors alone. Rather, they are attributable to the semantic alignment provided by CLIP-derived anchors, which consistently preserve accuracy under domain shifts, while disrupting or randomizing anchor semantics leads to large degradations.
>
> **Newer Baseline (W4, RC2):** Our current baselines follow those used in a recent paper specifically investigating domain generalization under noisy labels, which is why they are limited to earlier methods. We agree that including more recent approaches would strengthen the comparison, and we have added results for URM (2024) as a representative method. URM is especially relevant because it directly addresses a recently introduced problem of the interplay between domain shifts and label noise, making it a closer point of comparison than methods developed under the standard clean-label DG assumption. These additions help highlight that A3W achieves consistent improvements even against stronger recent baselines.
>
> **Paper Clarity (W5, RC1, RC3):** We have revised the manuscript accordingly:
> Noise accuracy (Fig. 1d): Now explicitly defined in Section 3.2 as accuracy w.r.t. corrupted labels, clarifying its role in measuring overfitting to noise.
> Equation 14: Added definitions for variables and explained the update schedule in words.
> Figures 4–6: Revised captions and axis labels to be more descriptive and self-contained.
>
> These details have been incorporated into the revised version. We have promptly updated the paper accordingly, and we remain committed to releasing the code upon acceptance as well as addressing any additional feedback in a timely manner.

---

### Review · Reviewer_46dc · 2025-09-06

**Summary Of Contributions:**

This paper proposes A3W, a method for domain generalization under label noise. It uses fixed class-level embeddings from a pretrained language-image model (e.g., CLIP) as semantic anchors, encouraging image features to align with them. Training samples are reweighted based on their cosine similarity to the anchors, downweighting likely noisy or spurious examples.

**Audience:**

Yes

**Broader Impact Concerns:**

The paper does not include a Broader Impact Statement, but given the technical nature and scope of the work, no major ethical concerns are identified.

**Claims And Evidence:**

Yes

**Requested Changes:**

1. Add stronger baselines
Please compare against recent, relevant methods:
DivideMix (ICLR 2020),TDG (2023)

2. Use more realistic noise
Current experiments use only symmetric noise. Please evaluate under instance-dependent or asymmetric label noise to better reflect real-world settings.

3. Validate CLIP anchor assumption
The method assumes that CLIP-derived anchors are domain-invariant. Please provide visualizations or analysis (e.g., cosine similarity plots, t-SNE) to verify this.

4. Clarify novelty vs TDG
The TDG paper uses similar CLIP-based guidance. Please clarify how A3W differs, and ideally include a comparison.

5. Discuss scalability concerns
A3W introduces class-specific projection heads. The authors should discuss how the method scales with large label spaces (e.g., ImageNet) and whether anchor sharing, clustering, or low-rank approximations could help.

6. Add ablation with random or perturbed anchors
To further demonstrate the value of CLIP-derived anchors, an ablation where anchors are replaced with random vectors or shuffled class embeddings would help verify that semantic structure is essential.

7. Improve theoretical justification
The current lemmas and theorems rely on intuitive arguments around gradient magnitude and softmax reweighting. Consider extending the analysis to connect with domain generalization theory (e.g., upper bounds involving $\mathcal{H}$-divergence or discrepancy distances).

**Strengths And Weaknesses:**

**Pros:**

1. Clear motivation: Combines language-guided semantics with noise-aware training.

2. Method is simple and easy to implement.

3. Consistent improvements over ERM, IRM, Mixup on several benchmarks.

**Cons / Limitations"**

1. Low novelty: Mostly a combination of CLIP embeddings and softmax reweighting; lacks new algorithmic insight.

2. Unverified assumptions: Assumes CLIP anchors are domain-invariant but provides no analysis or validation.

3. Weak theoretical support: Lemmas restate basic softmax behavior; no generalization bounds or new theoretical contributions.

4. Limited noise model: Only symmetric noise is used; no real-world or instance-dependent corruption.

5. Small ablation gains: Removing anchor or weighting drops accuracy by only ~1%, suggesting weak impact.

6. Scalability concern: One projection head per class may not scale to large-label datasets.

7. Missing strong baselines: No comparison to recent DG or noise-robust methods

---

> ### Author Response · Authors · 2025-09-20
>
> We thank the reviewer for the insightful comments and constructive suggestions. Below are our replies:
>
> **Low Novelty (W1, W3, RC7):** Our goal in this work is not to propose an entirely new algorithm, but to study a new setting of the interplay between label noise and domain generalization, which is a problem that has received little attention so far. For this reason, we build on established components like CLIP embeddings and softmax reweighting, adapting them to this context to isolate the role of noisy labels and test whether semantic anchors can improve robustness. We have also strengthened Section 3.8 with added intuition and formal bounds, showing how our approach acts as a soft regularizer that mitigates noise while reducing domain discrepancy. In this way, our contribution lies in framing and analyzing the new scenario, adapting existing mechanisms, and demonstrating both empirically and theoretically that these adaptations yield clear gains.
>
> **Domain Invariance of CLIP anchors (W2, W3, RC3):** In our method, we have the same prompt across different domains, and the heart of our method lies in the same embedding anchor across all domains, but variation in embedding anchor across different classes. So in short our CLIP anchors are the same across domains. The added ablation study further supports this point. Replacing anchors with random or shuffled vectors leads to clear performance drops (e.g., PACS 82.1% → 73.5% with random anchors, 77.2% with shuffled anchors), showing that semantic consistency across domains is key. The model still learns some discriminative features without meaningful anchors, but alignment across domains weakens and generalization suffers. By contrast, our full model achieves the best results overall, confirming that domain-invariant anchors are an essential part of the robustness we observe under noisy labels and domain shifts.
>
> **Realistic Noise Experiment (W4, RC2):** We agree and have added experiments on PACS with asymmetric label noise, where labels are flipped to confusable classes. Results at noise rates 0.1, 0.2, and 0.3 show that while Mixup degrades sharply (e.g., 78.7% at 0.3), our method remains robust (90.3% at 0.3). This confirms our gains extend beyond symmetric noise to more realistic corruption.
>
> **Scalability Concern (W6, RC5):** We agree that class-specific projection heads may become costly in large label spaces (e.g., ImageNet). In our current benchmarks, the overhead is modest (~10%), but we recognize that scaling requires additional strategies. As discussed in the revised manuscript, potential solutions include anchor sharing among related classes, clustering of NLP anchors, low-rank approximations of projection matrices, and hierarchical anchoring. We believe these extensions can preserve the benefits of semantic alignment while significantly improving scalability.
>
> **Stronger Baseline & vs. TDG (W7, RC1, RC4):** In response, we have expanded our comparison by adding results for URM (Krishnamachari et al., 2024) as a strong DG baseline, and we have reported these in the revised manuscript. Regarding DivideMix (Li et al., 2020), while this method is a seminal approach for learning with noisy labels, it is primarily designed for the single-domain noisy-label setting and does not directly address domain generalization. We therefore position DivideMix as complementary rather than directly comparable.
> As for TDG (Liu & Wang, 2023), we acknowledge the conceptual similarity since both TDG and A3W incorporate CLIP-based textual guidance. However, the two methods differ in important ways. TDG focuses on using text-guided domain-level augmentation and alignment to mitigate domain shift, whereas A3W introduces class-specific anchors and adaptive weighting to jointly handle domain shift and noisy labels. In particular, A3W dynamically down-weights misaligned or noisy samples, while TDG does not explicitly address label noise. Unfortunately, we were unable to locate publicly available code for TDG, which prevented a direct empirical comparison. We have nonetheless added a detailed discussion of TDG in our Related Work section and clarified how our contributions differ.
>
> (Continued on next comment)

---

> ### Author Response · Authors · 2025-09-20
>
> **Additional Ablation (W5, RC6):** Following the recommendation, we have added ablations in which the semantic anchors are replaced with either random vectors or shuffled class embeddings. Both variants preserve the training pipeline but disrupt the semantic structure that CLIP-derived anchors provide. The results confirm that semantic structure is essential: performance drops sharply when anchors are randomized or shuffled. For instance, on VLCS accuracy falls from 76.1% with our method to 47.5% with random anchors and 65.6% with shuffled anchors, while on DomainNet it decreases from 86.5% to 60.5% and 70.7%, respectively. Similar though smaller degradations appear on PACS, Office-Home, and SVIRO. These findings strengthen our claim that the gains do not stem from the mere presence of auxiliary vectors but rather from the meaningful semantic alignment of CLIP-derived anchors, which consistently supports robust generalization across domains.
>
> We have promptly incorporated these details into the revised manuscript. We also remain committed to open-sourcing the code upon acceptance, and will continue to address any additional comments in a timely manner.

---

### Comment · Reviewer_r5E5 · 2025-07-18
**Non-responsive action editor**

I asked the action editor twice (2025-06-17, 2025-06-25) to remove me from this assignment due to my lack of expertise in this area. There has been no response from the action editor. I am afraid I could not acknowledge this assignment, or post any review. Sorry about the inconvenience.

---

### Decision · Action_Editor_MAom · 2025-11-08

**Recommendation:** Reject

**Audience:**

Yes

**Audience Explanation:**

While the paper falls short of acceptance criteria, the core problem it addresses—jointly handling domain shift and label noise—represents a practically relevant challenge for the TMLR community. The proposed approach of leveraging fixed semantic anchors from vision-language models like CLIP provides a computationally efficient alternative to more complex domain generalization methods.

**Claims And Evidence:**

No

**Claims Explanation:**

While the authors have endeavored to address the reviewers' critiques in their rebuttal, the central claims of the submission still lack the necessary accuracy, convincing support, and evidentiary clarity required for publication. Reviewer 46dc acknowledges the paper's clear presentation and that some technical points were clarified, but concludes that this paper falls short of venue expectations. More critically, the lack of publicly available code fundamentally undermines the reproducibility of all experimental results, making independent verification of the authors' claims impossible.

Furthermore, the fundamental concerns raised by Reviewer iJEw regarding theoretical grounding, architectural justification, and experimental rigor remain largely unresolved. The reviewer notes that the theoretical assurances of robustness to label noise are not substantiated by formal guarantees, such as bounded gradients or a quantified analysis of the noise parameter's influence. The core architectural choice of domain-invariant label anchors is also critically challenged for its failure to incorporate valuable domain-specific style information—a limitation that the authors' response does not adequately justify. Additionally, the experimental setup raises serious concerns about reproducibility and fairness; baseline results are adopted from a prior work without including its method for direct comparison, and the stochastic label noise injection protocol introduces unquantified variability, especially for imbalanced datasets. Unfortunately, the authors' responses have not sufficiently alleviated these doubts regarding the consistency and reliability of the reported results.

**Resubmission Of Major Revision:**

The authors may consider submitting a major revision at a later time.